# Assessment of Functional Capacity in Patients with Nondialysis-Dependent Chronic Kidney Disease with the Glittre Activities of Daily Living Test

**DOI:** 10.3390/healthcare11121809

**Published:** 2023-06-20

**Authors:** Mauro Ribeiro Balata, Arthur Sá Ferreira, Ariane da Silva Sousa, Laura Felipe Meinertz, Luciana Milhomem de Sá, Vinicius Guterres Araujo, Jannis Papathanasiou, Agnaldo José Lopes

**Affiliations:** 1Postgraduate Program of Rehabilitation Sciences, Centro Universitário Augusto Motta (UNISUAM), Rio de Janeiro 21032-060, RJ, Brazil; m_balata@yahoo.com.br (M.R.B.); arthurde@souunisuam.com.br (A.S.F.); 2Medicine Course, Ceuma University (UNICEUMA), São Luís 65075-120, MA, Brazil; arisousa2@hotmail.com (A.d.S.S.); meinertzlf@gmail.com (L.F.M.); lucianamilhomem3@gmail.com (L.M.d.S.); 3State Hospital of High Complexity Dr. Carlos Macieira, São Luís 65070-220, MA, Brazil; viniciussguterress@gmail.com; 4Intensive Care Hospital, São Luís 65071-383, MA, Brazil; 5Department of Medical Imaging, Allergology & Physiotherapy, Faculty of Dental Medicine, Medical University of Plovdiv, 4002 Plovdiv, Bulgaria; giannipap@yahoo.co.uk; 6Department of Kinesitherapy, Faculty of Public Health “Prof. Dr. Tzecomir Vodenicharov, DSc.”, Medical University of Sofia, 1431 Sofia, Bulgaria; 7Postgraduate Program of Medical Sciences, School of Medical Sciences, State University of Rio de Janeiro (UERJ), Rio de Janeiro 20550-170, RJ, Brazil

**Keywords:** nondialysis-dependent chronic kidney disease, exercise, muscle, physical activity, health-related quality of life

## Abstract

This study evaluated the functional capacity measured by the Glittre-ADL test (TGlittre) in patients with nondialysis-dependent chronic kidney disease (NDD-CKD) and analyzed the test’s associations with muscle strength, physical activity level (PAL), and quality of life. Methods: Thirty patients with NDD-CKD underwent the following evaluations: the TGlittre; the International Physical Activity Questionnaire (IPAQ); the Short Form-36 (SF-36); and handgrip strength (HGS). The absolute value and percentage of the theoretical TGlittre time were 4.3 (3.3–5.2) min and 143.3 ± 32.7%, respectively. The main difficulties in completing the TGlittre were squatting to perform shelving and manual tasks, which were reported by 20% and 16.7% of participants, respectively. The TGlittre time correlated negatively with HGS (*r* = −0.513, *p* = 0.003). The TGlittre time was significantly different between the PALs considered “sedentary”, “irregularly active”, and “active” (*p* = 0.038). There were no significant correlations between TGlittre time and the SF-36 dimensions. Patients with NDD-CKD had a reduced functional capacity to exercise with difficulties performing squatting and manual tasks. There was a relationship between TGlittre time and both HGS and PAL. Thus, the incorporation of the TGlittre in the evaluation of these patients may improve the risk stratification and individualization of therapeutic care.

## 1. Introduction

Chronic kidney disease (CKD) consists of renal injury with progressive loss of kidney function, including glomerular, tubular, and endocrine functions [1]. CKD originates in heterogeneous pathways that irreversibly alter the function and structure of the kidneys over months or years. CKD affects a substantial proportion of the population, and its prevalence is increasing rapidly due to increased population ageing and the prevalence of diabetes, obesity, hypertension, and cardiovascular diseases that contribute to CKD [1]. It is estimated that 14% of the world’s population has CKD, and although 80% of cases are in late stages of the disease, it is likely that the actual proportion of people with early-stage CKD is much higher, since initial kidney disease is silent [2]. In addition to cardiovascular changes, individuals with CKD have muscle atrophy, anemia, fatigue, and cramps as consequences of the disease [3]. Overall, patients with CKD have low physical capacity and often struggle to complete basic activities of daily living (ADLs) [1,3].

It has already been widely discussed in the literature that patients with CKD commonly present loss of muscle mass, muscle weakness, and deterioration of physical function [4]. An important reduction in both muscle quality and function is observed in patients with CKD [5]. The prevalence of sarcopenia among patients with nondialysis-dependent CKD (NDD-CKD) can range from 5 to 60% and is associated with physical limitations and higher hospitalization rates compared to the general population [6,7]. Several factors can negatively affect the musculoskeletal system in this population, such as changes in muscle perfusion, an imbalance between anabolism and catabolism, reduced protein intake, insulin resistance, the presence of metabolic acidosis, the use of corticosteroids, and the presence of pro-inflammatory cytokines [6,8]. These factors can potentially lead to poor performance in physical activities and reduced functional capacity during exercise [9]. As a consequence, there may be a deterioration in physical mobility that contributes to a worsened general health status and increased mortality [10]. Considering the multiple aspects involving CKD patients, the functional capacity to exercise should be measured through tasks that represent ADLs more broadly, rather than tests directed at isolated ADL components such as the six-minute walk test.

Patients with NDD-CKD may benefit from a detailed evaluation of physical capacity if a more functional approach is also taken into account. Functional tests may be even more advantageous if they are able to reflect the functional performance represented by ADLs. In recent years, the Glittre-ADL test (TGlittre) has emerged as an instrument for assessing functional capacity to exercise by the performance of multiple ADLs incorporating the muscle activity of the upper and lower limbs. The TGlittre is a reliable and valid instrument for functional evaluation in different populations and has even been used among patients undergoing hemodialysis [9]. Since an increasing number of patients have NDD-CKD with the potential to achieve better physical health outcomes than with the dialysis route [11], it is essential to evaluate the functional capacity to exercise in this patient population. The present study aimed to evaluate the functional capacity to exercise measured by the TGlittre among patients with NDD-CKD and to analyze its associations with muscle strength, physical activity level (PAL), and health-related quality of life (HRQoL).

## 2. Materials and Methods

### 2.1. Study Design and Participants

Between January and June 2022, a cross-sectional and quasi-experimental study was conducted among 30 (of 33 eligible) patients with NDD-CKD who were followed up as outpatients at the Vila Esperança Specialty Center, São Luís, MA, Brazil. The patients were aged ≥18 years with proven NDD-CKD for a period >3 months by an estimated glomerular filtration rate (eGFR) ≤60 mL/min per 1.73 m^2^ or albuminuria >30 mg/dL in an isolated sample or 24-h urine [12]. The following exclusion criteria were used: recent myocardial infarction; malignant ventricular arrhythmias; unstable angina; systolic blood pressure >200 mmHg and diastolic blood pressure >120 mmHg at rest; decompensated diabetes; chronic lung diseases; and inability to walk independently or need for any assistive device.

The protocol was approved by the Research Ethics Committee of the Augusto Motta University Centre, Rio de Janeiro, Brazil, under number 52697821.5.0000.5235. This study was conducted according to the principles of the Declaration of Helsinki. All patients signed an informed consent form.

### 2.2. Chronic Kidney Disease Risk Categories

According to the “Kidney Disease: Improving Global Outcomes (KDIGO) 2012” guidelines [12], participants were classified into six eGFR categories and three albuminuria categories (Figure 1). The eGFR is the result of a calculation obtained from the measurement of serum creatinine and the patient’s age and sex [13]. Through the combined assessment of eGFR and the urine-albumin–creatinine ratio (uACR), a patient can be more accurately evaluated as being at low, moderately increased, high, or very high risk of worsening kidney function and other complications [14].

### 2.3. Assessment of Physical Activity Level

PAL in daily life was evaluated by the short form of the International Physical Activity Questionnaire (IPAQ) [15]. The IPAQ consists of eight open-ended questions that evaluate the time and frequency of walking and moderate and vigorous activities in the past week to evaluate the time spent per week engaged in physical activity. ADLs were divided into different intensities (mild, moderate, and vigorous) for the following domains: work; means of transport; household chores; recreational activities, sports, physical exercise, and leisure; and time spent in passive activities performed in a sitting position.

### 2.4. Assessment of Health-Related Quality of Life

The Short Form-36 (SF-36) was used to evaluate HRQoL. This is a multidimensional and self-administered tool composed of 36 items grouped into 8 dimensions: physical functioning, physical role limitations, bodily pain, general health perceptions, vitality, social functioning, emotional role limitations, and mental health [16]. The results are measured in scores from 0 to 100, and the higher the score is, the better the HRQoL.

### 2.5. Assessment of Handgrip Strength

For the evaluation of handgrip strength (HGS), a digital handheld dynamometer (SH5001, Saehan Corporation, YangdeokDong, Masan, Korea) was used. Participants were instructed to perform a maximum contraction for 3 s in each test. Three measurements with an interval of 30 s were performed. The participants were comfortably seated in an armless chair with their feet on the floor and their hips and knees positioned at approximately 90 degrees of flexion. The shoulder of the tested limb was adducted and in neutral rotation, the elbow was flexed at 90 degrees, the forearm was in a neutral position, and the wrist was between 0 and 30 degrees of extension and between 0 and 15 degrees of adduction. The hand of the nontested limb rested on the thigh on the same side. The highest value was considered for analysis [17]. For comparative purposes, the Brazilian predicted values for healthy individuals were used [18]. HGS was classified as low if values less than 32 kg were obtained for men and less than 17 kg for women [19].

### 2.6. Assessment of Functional Capacity

The TGlittre was performed as previously proposed [20] (Figure 2). In the present study, the protocol was performed twice with an interval of 30 min, and the shorter TGlittre time was used for analysis [21]. For comparative purposes, the Brazilian predicted values for healthy individuals were used [22].

### 2.7. Statistical Methods

The normality of the distribution of the variables was assessed by the Shapiro-Wilk test. The correlations of TGlittre time with anthropometric data, HRQoL, muscle strength, comorbidities, risk of disease progression, and PAL were analyzed with Pearson’s (*r*) or Spearman’s (*r_s_*) correlation coefficients for numerical variables and by Student’s *t* test for independent samples or one-way analysis of variance (ANOVA) for categorical variables. Additionally, Tukey’s multiple comparison test was used to identify the subgroups that differed significantly from each other. For comparison purposes, participants were grouped into low/moderate, high, or very high risk according to the KDIGO 2012 guidelines [12]. A generalized linear model was used to identify independent numerical and categorical predictors for TGlittre time. The results are expressed as the means ± standard deviations (SD), medians (interquartile ranges) or frequencies (percentages). Statistical significance was defined as *p* < 0.05. Data analysis was performed using the statistical software SPSS version 26.0 for Windows.

## 3. Results

Among the 33 patients with NDD-CKD who were evaluated for inclusion in this study, 3 were excluded for the following reasons: 2 patients had walking difficulties and 1 patient had suffered a recent myocardial infarction. The mean age was 58.1 ± 13.9 years, while the median time after diagnosis of CKD was 18 (2–42) months. The main causes of CKD were hypertension, diabetes, and chronic and idiopathic glomerulopathy, which occurred in 23 (76.7%), 12 (40%), 5 (16.7%), and 3 (10%) participants, respectively. Almost half of the participants (14, 46.7% of cases) had a very high risk of disease progression. Regarding the SF-36, the dimensions with the worst scores were physical role limitations and general health perceptions, with median scores of 50 (19–100) and 60 (42–67), respectively. Anthropometric data, comorbidities, renal function, risk of disease progression, and HRQoL are shown in Table 1.

Regarding PAL, almost half of the participants (13, 43.3%) were considered “sedentary” by the IPAQ. Regarding the evaluation of peripheral muscle strength, the median absolute value of HGS was 29 (26–41) kgf, while the median absolute value in relation to the theoretical value for the Brazilian population in the study by Neves et al. [18] was 78 (60–91%) predicted. HGS was considered normal and reduced in 8 (26.7%) and 22 (73.3%) participants, respectively. Regarding the TGlittre, the median time to perform the activities was 4.3 (3.3–5.2) min, with a mean predicted value of 143.3 ± 32.7%. The main difficulties in completing the TGlittre were squatting to perform shelving tasks and manual tasks, which were reported by 6 (20%) and 5 (16.7%) participants, respectively. PAL, muscle strength, and functional capacity data are shown in Table 2.

The associations between TGlittre time and anthropometric data, HRQoL, and muscle function data are shown in Table 3 and Figure 3. The TGlittre time was negatively correlated with HGS (*r_s_* = −0.513, *p* = 0.003). We did not observe significant correlations between TGlittre time and the dimensions of the SF-36. The associations between TGlittre time and the data related to sex, comorbidities, risk of disease progression, and PAL are shown in Table 4 and Figure 4. In this analysis, the TGlittre time was significantly different between the PALs considered “sedentary”, “irregularly active”, and “active” (160 ± 22 s vs. 146 ± 40 s vs. 122 ± 26 s, respectively, *p* = 0.038). The TGlittre time was significantly different between the participants who reported no difficulty and those who reported any difficulty in performing the TGlittre tasks (134 ± 28 s vs. 151 ± 34 s, respectively, *p* = 0.003). Notably, the TGlittre time was not significantly different between the participants in relation to the risk of disease progression according to KDIGO 2012 (147 ± 48 s vs. 139 ± 28 s vs. 151 ± 28 s for low/moderate risk, high risk, and very high risk, respectively, *p* = 0.69).

Table 5 shows the multivariate analysis according to the generalized linear models. In this analysis, HGS (*p* = 0.004) and the IPAQ classification “active” (*p* = 0.014) were the only significant independent variables that explained TGlittre time; the other variables showed no independent contribution at the 5% level, including the risk of disease progression according to KDIGO 2012 and comorbidities.

## 4. Discussion

Functional tests seek to represent ADLs as accurately as possible to improve the evaluation and make it more reliable in the clinical environment. In this sense, the TGlittre is quite accurate in portraying ADLs because, in addition to simulating walking activities, it includes walking up and down steps, sitting and standing, and trunk and upper limb movements with load [20]. The main findings of the present study were that patients with NDD-CKD spent a great deal of time performing TGlittre tasks, and their execution was hampered mainly by squatting to perform shelving tasks and manual tasks. In these patients, TGlittre time was associated with both HGS and PAL, which were independent performance variables during the test. To our knowledge, this is the first study to evaluate the performance of patients with NDD-CKD using the TGlittre.

In our study, we observed that the median TGlittre time was approximately 43% longer than predicted for the Brazilian population. The underlying mechanisms for the reduction in functional capacity to exercise in these patients are multifactorial and may be associated with impaired renal function leading to anemia, uremic neuropathy, myopathy, and cardiovascular abnormalities [4]. Interestingly, we observed that in absolute values, patients with NDD-CKD spent approximately 4.3 min completing the TGlittre, which is a value well below that observed among patients on hemodialysis (4.8 min, 95% confidence interval (CI) 4.4–5.3) in the study by Figueiredo et al. [9]. This helps to corroborate our hypothesis that among patients on dialysis, the burden of kidney disease is greater than that among patients with NDD-CKD, substantially affecting their functional capacity to exercise [11]. Notably, we did not observe any association between TGlittre time and NDD-CKD-related comorbidities (e.g., diabetes and hypertension), which suggests that the severity of kidney disease is a more important factor in reducing the functional capacity to exercise than the cause of NDD-CKD [4].

CKD has recently been defined as a “model of accelerated ageing”, influencing the human body in a way that is patently comparable to ageing [5]. In our study, more than 70% of patients had reduced HGS, which is an important index in the diagnosis of sarcopenia, a robust predictor of low muscle mass, and a clinical marker of low physical performance [23]. In patients with CKD, the reduction in muscle strength becomes apparent from the onset of the disease and is multifactorial, since CKD contributes to a catabolic state due to increased muscle proteolysis and reduced protein synthesis. This reduction in muscle mass leads to a sedentary lifestyle, exercise intolerance, and low cardiorespiratory fitness, which causes functional limitations and increased mortality [7,24,25]. In fact, HGS was the main determinant of TGlittre performance in our multiple regression model. The association between HGS and TGlittre performance has also been described among patients on hemodialysis [9] and may have contributed to the difficulty in performing the shelf tasks in our sample. Thus, muscle strength measures, which can be easily evaluated by HGS, should be incorporated as an important component for the diagnosis of muscle disorders in people with CKD [7]. Because low bone mineral density is prevalent and associated with low markers of muscle mass and quality in patients with NDD-CKD, targeted interventions are needed to optimize the body composition and functional status of these patients [26].

Functional disability in patients with CKD is multifactorial and is associated with cardiovascular disease, muscle weakness, reduced eGFR, and sedentary behavior [27]. In fact, we observed an association between TGlittre time and PAL. Although no previous studies have used the TGlittre among patients with NDD-CKD, a recent study that used the TGlittre among hemodialysis patients showed a moderate correlation between TGlittre performance and moderate to vigorous physical activity evaluated by accelerometry [4]. According to these researchers, the reduced TGlittre performance observed among hemodialysis patients can be justified not only by the weekly sedentary period imposed by dialysis treatment but also by CKD-related osteosarcopenia, which negatively affects mobility and ADLs. In evaluating hemodialysis patients, Figueiredo et al. [9] observed a correlation between TGlittre performance and patients’ ADLs, suggesting that the faster the patients complete the TGlittre, the higher their PALs. Despite lower adherence, patients with CKD show clinically significant benefits from physical activity, with no apparent impact on safety, compared to those without CKD [28].

The general population has a better HRQoL than patients with CKD of all stages, although patients with NDD-CKD or kidney transplant patients have a better HRQoL than patients on dialysis [29]. There are many factors that have a negative impact on the HRQoL of patients with CKD, including depression, anxiety, cognitive deficit, inactivity and fragility in the physical domain, as well as a lack of social support [29]. Although we observed a low HRQoL evaluated by the SF–36, no significant correlation was observed with the TGlittre. Unlike our results, however, TGlittre performance has been associated with a worse HRQoL among patients on hemodialysis, especially in relation to the physical domains [9]. A possible explanation for this discrepancy may be the lower severity of NDD-CKD patients, who experience a smaller impact on physical measures, such as strength, resistance, and mobility.

Some limitations of our study should be mentioned. First, the small sample size and nonrandomized nature of this study may limit the possibility of generalizing our findings. Accordingly, our study did not include a control group of healthy individuals, although the discrepancies are possibly enormous in relation to patients with NDD-CKD. Furthermore, as this study used a quasi-experimental design, there was no prediction of comparison with a control group [30]. Third, we did not use motion sensors as accelerometers that could more objectively assess physical activity, since PAL seems to strongly influence patients with CKD. Despite these limitations, our results can be used to guide researchers and rehabilitation professionals in the interpretation of functional changes detected in patients with NDD-CKD and in clinical trials.

## 5. Conclusions

As evaluated by the TGlittre, patients with NDD-CKD had a reduced functional capacity to exercise with difficulties in performing squatting and manual tasks. In addition, there was a relationship between TGlittre time and both HGS and PAL. Thus, the incorporation of the TGlittre into the routine evaluation of patients with NDD-CKD may improve the risk stratification and individualization of therapeutic care.

## Figures and Tables

**Figure 1 healthcare-11-01809-f001:**
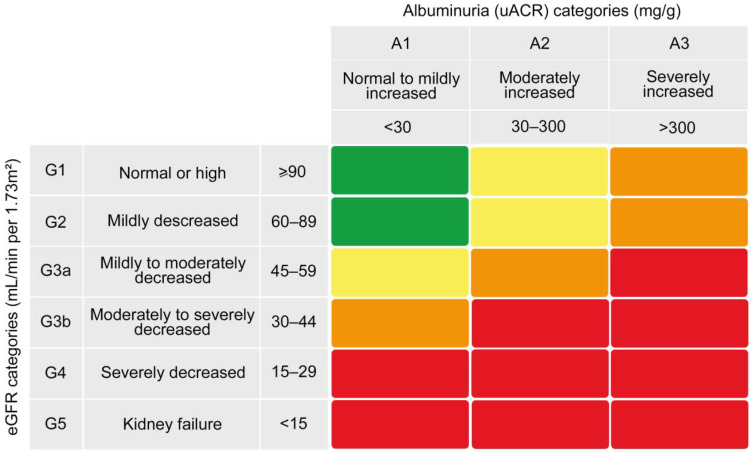
Prognosis of chronic kidney disease by estimated glomerular filtration rate (eGFR) and albuminuria categories. Green, low risk of disease progression; yellow, moderately increased risk of disease progression; orange, high risk of disease progression; and red, very high risk of disease progression. uACR: urine-albumin–creatinine ratio.

**Figure 2 healthcare-11-01809-f002:**
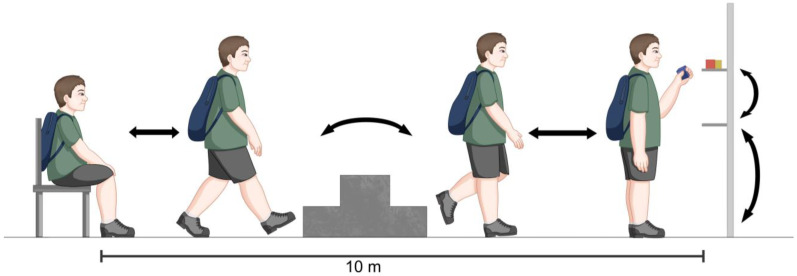
Schematic representation of the Glittre-ADL test. The test consists of a standardized circuit in which the individual is instructed to go through the following sequence of activities in the shortest possible time: sitting in front of a 10 m course, the individual stands up and walks on the plane. In the middle of the circuit, they go up and down two steps and walk again on the plane. At the end of the circuit, there is a shelf on which the individual must move three objects weighing 1 kg each positioned on the highest shelf, one by one, to the lowest shelf and then to the floor. The objects must be placed again on the lowest shelf and finally return to the highest shelf. Then, the individual returns, taking the opposite route. For the test to be considered complete, the individual must perform five laps without any verbal encouragement. During the test, the individual wears a backpack weighing 2.5 kg for women and 5 kg for men.

**Figure 3 healthcare-11-01809-f003:**
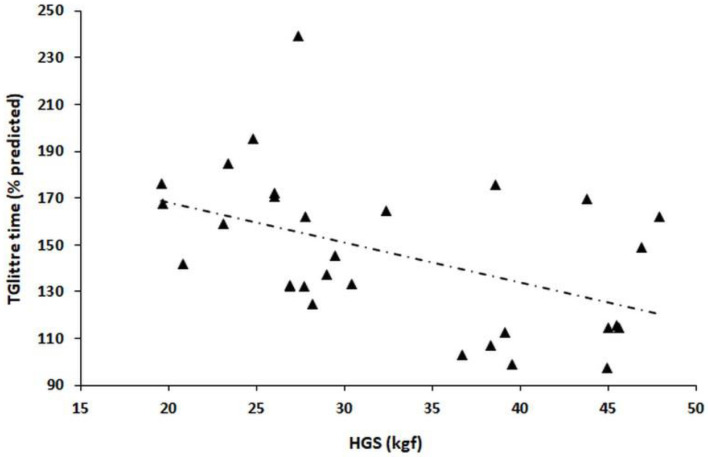
Relationship of Glittre-ADL test (TGlittre) time with handgrip strength (HGS, *r_s_* = −0.513, *p* = 0.003).

**Figure 4 healthcare-11-01809-f004:**
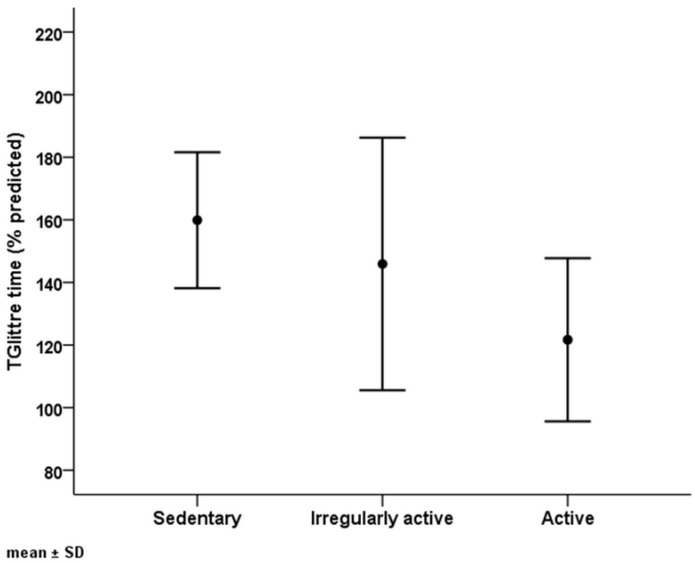
Glittre–ADL test (TGlittre) time values according to physical activity level.

**Table 1 healthcare-11-01809-t001:** Anthropometry data, comorbidities, renal function, risk of disease progression, and health-related quality of life in the studied sample.

Variable	Values
**Anthropometry**	
Male/female	19/11
Age (years)	58.1 ± 13.9
Weight (kg)	69.5 ± 10.6
Height (m)	1.63 ± 0.08
BMI (kg/m^2^)	26.2 ± 3.8
**Comorbidities, *n*** (**%**)	
Hypertension	23 (76.7)
Diabetes	12 (40)
Chronic heart disease	8 (26.7)
Chronic glomerulopathy	5 (16.7)
Chronic vascular disease	5 (16.7)
**Renal function**	
eGFR (mL/min)	39.9 ± 22
uACR (mg/g)	489.3 ± 78.2
**Risk of disease progression, *n*** (**%**)	
Low/moderate	7 (23.3)
High	9 (30)
Very high	14 (46.7)
**SF-36**	
Physical functioning (score)	80 (54–95)
Physical role limitations (score)	50 (19–100)
Bodily pain (score)	62 (41–100)
General health perceptions (score)	60 (42–67)
Vitality (score)	68 (55–90)
Social functioning (score)	81 (50–100)
Emotional role limitations (score)	100 (0–100)
Mental health (score)	76 (60–92)

The results are expressed as mean ± SD, median (interquartile range) or number (%); BMI: body mass index; eGFR: estimated glomerular filtration rate; uACR: urine-albumin–creatinin ratio; SF-36: Short Form-36.

**Table 2 healthcare-11-01809-t002:** Physical activity level, muscle function, and functional capacity in the studied sample.

Variable	Values
**IPAQ stages, *n*** (**%**)	
Sedentary	13 (43.3)
Irregularly active	10 (33.3)
Active	7 (23.3)
**Muscle function**	
HGS (kgf)	29 (26–41)
**Glittre-ADL test**	
Time (min)	4.3 (3.3–5.2)
Time (% predicted)	143.3 ± 32.7
Highest-difficulty task, *n* (%)	
No difficulty	16 (53.3)
Squatting to perform shelving tasks	6 (20)
Manual tasks	5 (16.7)
Stair tasks	3 (10)

The results are expressed as median (interquartile range) or number (%); HGS: handgrip strength; IPAQ: International Physical Activity Questionnaire.

**Table 3 healthcare-11-01809-t003:** Correlation coefficients for Glittre-ADL test, anthropometry, health-related quality of life, and muscle function among patients with nondialysis-dependent chronic kidney disease *.

Variable	Time (% Predicted)	
	** *r* **	** *p* ** **-value**
Age	0.296	0.11
Weight	−0.224	0.23
Height	0.254	0.17
BMI	−0.075	0.69
	*r_s_*	** *p* ** **-value**
Physical functioning	−0.217	0.25
Physical role limitations	−0.246	0.19
Bodily pain	−0.219	0.25
General health perceptions	0.183	0.33
Vitality	−0.098	0.61
Social functioning	−0.098	0.61
Emotional role limitations	−0.228	0.23
Mental health	−0.074	0.70
HGS	−0.513	**0.003**

* Appropriate Pearson’s (*r*) or Spearman’s (*r_s_*) correlation coefficients for variables with normal or nonnormal distribution, respectively. BMI: body mass index; HGS: handgrip strength.

**Table 4 healthcare-11-01809-t004:** Associations of the Glittre-ADL test with gender, comorbidities, risk of disease progression, and physical activity level among patients with nondialysis-dependent chronic kidney disease. * Appropriate Pearson (or Spearman) correlation coefficients for variables with normal (or nonnormal) distribution.

Variable	Time (% Predicted)	*p*-Value
**Gender**		
Male	142 ± 38	0.36 *
Female	154 ± 21	
**Hypertension**		
Yes	148 ± 28	0.62 *
No	141 ± 48	
**Diabetes**		
Yes	147 ± 27	0.98 *
No	146 ± 37	
**Chronic heart disease**		
Yes	157 ± 24	0.31 *
No	143 ± 35	
**Glomerulopathy**		
Yes	138 ± 31	0.53 *
No	148 ± 33	
**Chronic vascular disease**		
Yes	141 ± 34	0.71 *
No	147 ± 33	
**Risk of disease progression**		
Low/moderate	147 ± 48	0.69 ^#^
High	139 ± 28	
Very high	151 ± 28	
**IPAQ stages**		
Sedentary	160 ± 22	**0.038** ^#^
Irregularly active	146 ± 40	
Active	122 ± 26	

IPAQ: International Physical Activity Questionnaire. * The *p*-value was calculated using the Student’s t test. # The *p*-value was calculated using ANOVA with corrections by Tukey’s test (sedentary # active; sedentary = irregularly active; irregularly active = active).

**Table 5 healthcare-11-01809-t005:** Multiple linear regression according to generalized linear models for the Glittre-ADL test (TGlittre) time.

Independent Variable	Coefficient	Standard Error	*p*-Value
**HGS**	−1.367	0.477	**0.004**
**IPAQ stages**			
Sedentary	Reference		
Irregularly active	−15.9	12	0.18
Active	−28.1	11.5	**0.014**

HGS: handgrip strength; IPAQ: International Physical Activity Questionnaire.

## Data Availability

The dataset and scripts for statistical analysis can be obtained upon request to the principal investigator.

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
