# Peer review of "Assessment of Functional Capacity in Patients with Nondialysis-Dependent Chronic Kidney Disease with the Glittre Activities of Daily Living Test"

_healthcare, 2023, doi:10.3390/healthcare11121809_

Round 1

Reviewer 1 Report

This article evaluated of the functional capacity of patients with chronic kidney disease using the ADL-Glittre test and other tests. The total number was limited.

Major flaw

1 No normal control.

2 No CKD stage differentiation.

3 No adjustation of serum creatinine or other Comorbidities in Multiple linear regression.

4 Pearson's correlation coefficients were not suitable for unnormal distribution variables.

Reviewer 2 Report

In the manuscript entitled “Assessment of Functional Capacity in Patients with Nondialysis-Dependent Chronic Kidney Disease with the AVD-Glittre Test”, the authors employed Glittre-ADL test (TGlittre) to evaluate the functional capacity of nondialysis-dependent chronic kidney disease (NDD-CKD) patients and found that NDD-CKD patients have reduced functional capacity evaluated by TGlittre. At the same time, TGlittre time is related to handgrip strength and physical activity level. Performing TGlittre as routine evaluation of NDD-CKD patients can contribute to risk stratification and individual therapeutic care. However, there are some issues should be addressed:

Major Concerns:

1.     A schematic figure for TGlittre is recommended to depict how this test performs.

2.     The renal function index of the NDD-CKD patients in this study should be included.

3.     How to determine the risk level of disease progression should be illustrated.

Minor Concerns:

1.     The format of Figure 2 is incorrect. “{” is not common to show significant difference.

2.     English writing should be improved.

Round 2

Reviewer 1 Report

None.

Reviewer 2 Report

I have reviewed the revised manuscript and believe it is qualified to be published.